# Relationship between gut microbiota and circulating metabolites in population-based cohorts

Dina Vojinovic[1,6]*, Djawad Radjabzadeh[2,6], Alexander Kurilshikov [3,6], Najaf Amin[1], Cisca Wijmenga [3], Lude Franke[3], M. Arfan Ikram [1], Andre G. Uitterlinden [1,2], Alexandra Zhernakova[3], Jingyuan Fu[3,4,7], Robert Kraaij[2,7] & Cornelia M. van Duijn[1,5,7]*

Gut microbiota has been implicated in major diseases affecting the human population and has also been linked to triglycerides and high-density lipoprotein levels in the circulation. Recent development in metabolomics allows classifying the lipoprotein particles into more details. Here, we examine the impact of gut microbiota on circulating metabolites measured by Nuclear Magnetic Resonance technology in 2309 individuals from the Rotterdam Study and the LifeLines-DEEP cohort. We assess the relationship between gut microbiota and metabolites by linear regression analysis while adjusting for age, sex, body-mass index, technical covariates, medication use, and multiple testing. We report an association of 32 microbial families and genera with very-low-density and high-density subfractions, serum lipid measures, glycolysis-related metabolites, ketone bodies, amino acids, and acute-phase reaction markers. These observations provide insights into the role of microbiota in host metabolism and support the potential of gut microbiota as a target for therapeutic and preventive interventions.

[1] Department of Epidemiology, Erasmus MC, University Medical Center, Rotterdam, The Netherlands. [2] Department of Internal Medicine, Erasmus MC, University Medical Center, Rotterdam, The Netherlands. [3] University of Groningen, University Medical Center Groningen, Department of Genetics, Groningen, The Netherlands. [4] Department of Pediatrics, University of Groningen and University Medical Center Groningen, Groningen, The Netherlands. [5] Nuffield Department of Population Health, University of Oxford, Oxford, UK. [6] These authors contributed equally: Dina Vojinovic, Djawad Radjabzadeh, Alexander Kurilshikov. [7] These authors jointly supervised to this work: Jingyuan Fu, Robert Kraaij, Cornelia M. van Duijn. *email: d.vojinovic@erasmusmc.nl; Cornelia.vanDuijn@ndph.ox.ac.uk

There is increasing interest in the role of the gut microbiota in the major diseases affecting the human population. For a large part, these associations can be attributed to metabolic and immune signals of the microbiota that enter the circulation[1]. The gut microbiota has been implicated in obesity and diabetes[2], while recently it was also shown that the microbiota is also a substantial driver of circulating lipid levels, including triglycerides and high-density lipoproteins (HDL)[3–5]. The association with low-density lipoprotein (LDL) cholesterol levels, the major target for treatment of dyslipidemia, or total cholesterol was weaker than the association with triglycerides and HDL[3,4]. Recent development in metabolomics allows subclassifying the lipoprotein classes into more detail based on their particle size, composition, and concentration. Various studies further linked the gut microbiota to various amino acids, which have been implicated in diabetes and cardiovascular diseases[6–10].

To provide novel insights into the relation of gut microbiota and circulating metabolites, we perform an in-depth study of the metabolome characterized by nuclear magnetic resonance ($^1$H-NMR) technology in two large population-based prospective studies which have a rich amount of data on risk factors and disease. We identify 32 microbial families and genera associated with very-low-density and high-density subfractions, serum lipid measures, glycolysis-related metabolites, amino acids, and acute-phase reaction markers.

## Results

**Characteristics of study population.** Our study is embedded within the Rotterdam Study and LifeLines-DEEP cohort. The Rotterdam Study is a prospective population-based cohort study that started in 1990 among individuals from the well-defined district of Rotterdam[11], while LifeLines-DEEP cohort is a sub-cohort of LifeLines study, a prospective population-based cohort study in the north of the Netherlands established in 2006[12]. Participants from the Rotterdam Study ($n = 1390$, mean age $56.9 \pm 5.9$, 57.5% women) were older compared to the participants from LifeLines-DEEP study ($n = 915$, mean age $44 \pm 13.9$, 58.7% women), while sex distribution in the two cohorts was comparable.

**Association of gut microbiota with circulating metabolites.** The results of association analysis between circulating metabolites (Supplementary Data 1) and composition of gut microbiota (Supplementary Data 2) are illustrated in Fig. 1. There were 32 microbial families and genera associated with various circulating metabolites after adjusting for age, sex, body mass index (BMI), medication use, including lipid-lowering medication, protein-pump inhibitors, and metformin, technical covariates, and multiple testing (Fig. 1, Source Data). The variables corrected for in the analysis were selected according to previous literature findings[3,13]. The multiple testing correction included Bonferroni correction which was applied for the number of independent tests in both metabolomics and gut microbiota datasets calculated by a method of Li and Ji[14] ($0.05/(37$ independent metabolite measures $\times 274$ independent microbial taxa $= 4.93 \times 10^{-6}$). After additional adjustment for smoking and alcohol intake, similar association pattern was observed (Fig. 1b, Source Data). The direction of effect size across the cohorts was generally concordant (Supplementary Figs. 1–12 and Source Data).

We detected significant associations between 18 microbial families and genera and very-low-density (VLDL) particles of various sizes (extra small, small, medium, large, very large, and extremely large) and 22 microbial families and genera and HDL particles (small, medium, large, and very large) (Source Data). Abnormalities in VLDL particle distribution were reported to be associated with metabolic disease etiology, cardiovascular diseases, and type 2 diabetes[7,15–17], while inverse association of very large and large HDL particles and small and medium HDL particles was reported in relation to disease risk[7]. There were 13 microbial families and genera associated with both VLDL and HDL particles subclasses. For example, family *Christensenellaceae* and genera *Christensenellaceae R7 group*, *Ruminococcaceae* (*UCG-005*, *UCG-003*, *UCG-002*, and *UCG-010*), *Marvinbryantia* and *Lachnospiraceae FCS020 group* were found to be associated with VLDL particles of various sizes, small HDL particles and triglycerides in medium HDL particles (Source Data), whereas family *Clostridiaceae1* and genus *Clostridium sensu stricto 1* were additionally associated with very large and large HDL particles (Fig. 1b). Correlation analysis between these microbial taxa revealed positive correlation between family *Christensenellaceae*, *Christensenellaceae R7 group* genus, and *Ruminococcaceae* genera ($\rho$ ranged between 0.32 and 0.67) and within *Ruminococcaceae* genera ($\rho$ ranged between 0.46 and 0.77) (Supplementary Data 3). These correlations are not unexpected and were reported by previous studies[18,19]. Family *Christensenellaceae* was previously associated with BMI[20], genus *Marvinbryantia* was liked to bowel dysfunction[21], while family *Clostridiaceae1* is involved in bile acid metabolism and liked to BMI[4,22]. Of note is that the association pattern of very large and large HDL particles, including concentration of particles and its total lipids, cholesterol, free cholesterol, and cholesterol esters was opposite compared to the association pattern of small and medium HDL (Fig. 1).

In addition, we confirmed previously reported association of serum triglycerides and genus *Ruminococcus gnavus*, a gut microbe linked to low gut microbial richness[4,23]. There were 15 microbial families and genera associated with serum triglycerides (Fig. 1). The association pattern of serum triglycerides clustered with triglycerides in small VLDL, LDL and HDL, medium HDL, and cholesterol and cholesterol ester in medium VLDL (Supplementary Fig. 13). Of 15 microbial families and genera associated with serum triglycerides there were three microbial taxa were also associated with VLDL, LDL, and HDL particles and 9 were associated with VLDL and HDL particles (Supplementary Fig. 14).

We also identified an association between family *Lachnospiraceae* and its genus *Blautia* with small HDL particles (Fig. 1b). Genera from family *Lachnospiraceae*, one of the major taxonomic groups of the human gut microbiota, have been associated with the maintenance of gut health, and genus *Blautia* was associated with obesity and reported to be involved in conversion of primary bile acids into secondary bile acids[24–26]. Correlation coefficient between family *Lachnospiraceae* and genus *Blautia* that belongs to this family was 0.82 (Supplementary Data 3). Family *Clostridiaceae1* and genus *Clostridium sensu stricto 1* were associated with diameter of both HDL and VLDL particles (Fig. 1b). The VLDL diameter was further associated with family *Christensenellaceae* and genera *Christensenellaceae R7 group* and *Ruminococcaceae UCG-002*. Interestingly, diameter of HDL particles was linked to cardiovascular disease [7,8].

Beyond the lipoprotein fractions, six microbial families and genera were associated with fatty acids, including monounsaturated (MUFA), saturated (SFA), and total fatty acids (TotFA), while eight microbial families and genera were associated with three other metabolites, including the ketone body acetate, amino acid isoleucine, and acute-phase reaction marker glycoprotein acetyl (mainly alpha 1) (Fig. 1b, Source Data). Genus *Ruminococcaceae UCG-005* was associated with acetate, while family *Clostridiaceae1* and genera *Clostridium sensu stricto 1* and *Ruminococcaceae UCG*-014 showed association with isoleucine. Interestingly, seven microbial families and genera were associated with glycoprotein acetyls levels which are known to be associated

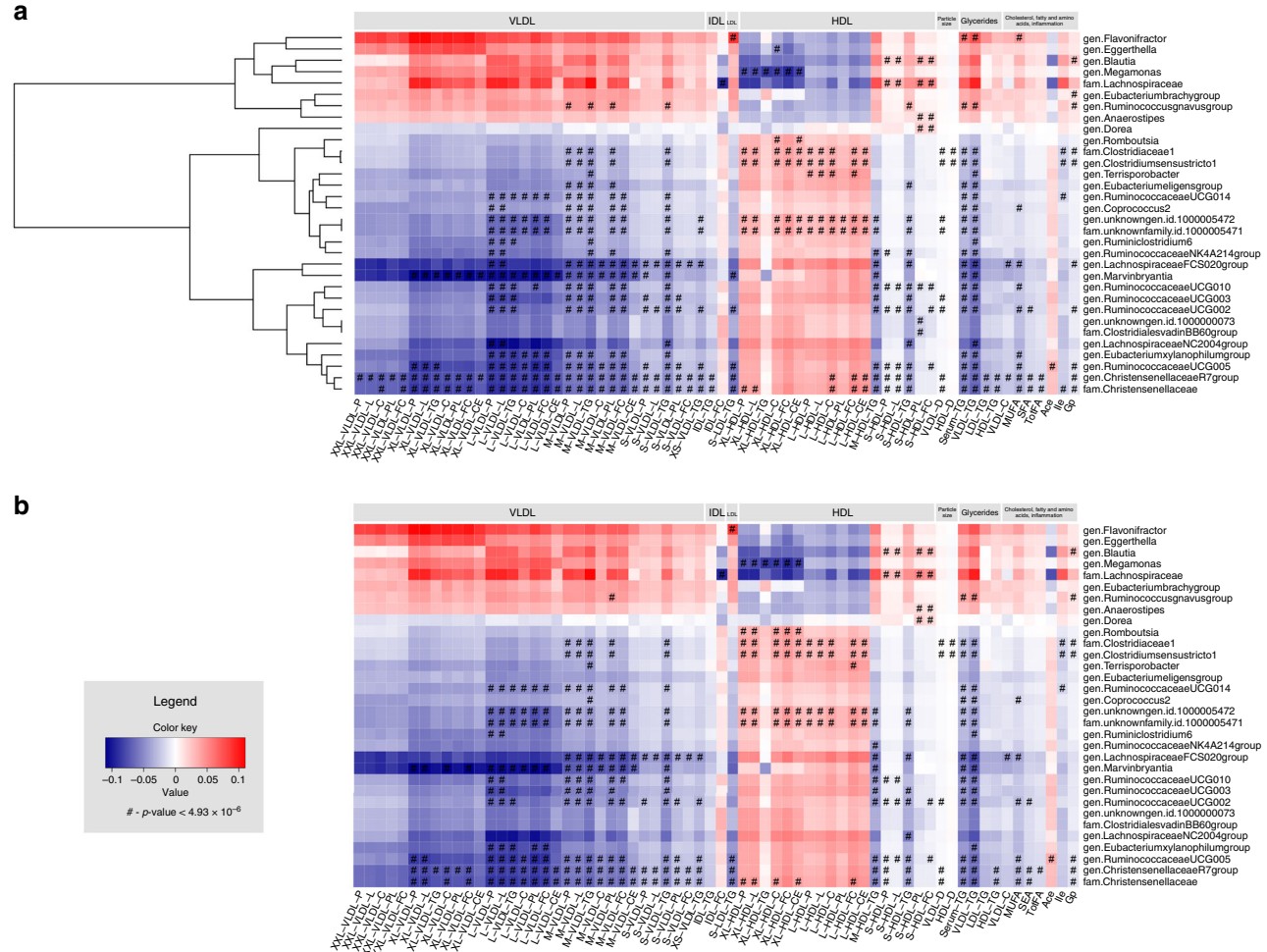

**Fig. 1 Results of association analysis between metabolites and microbial genera and families assessed by linear regression analysis ($n = 2309$).**
Association results after adjustment for age, sex, body mass index, technical covariates, and medication use are displayed on panel (**a**), while association results after additional adjustment for smoking and alcohol consumption are shown on panel (**b**). Metabolites are displayed on x-axis, whereas microbial genera and families are shown on y-axis. Lipoprotein classes include very-low-density lipoprotein particles (VLDL), intermediate lipoprotein particles (IDL), low-density lipoprotein particles (LDL), and high-density lipoprotein particles (HDL) of very low (XS), low (S), medium (M), large (L), very large (XL), and extremely large (XXL) size. Blue color stands for inverse association. Red color denotes positive associations. Symbols on the plot represent the level of significance with hash denoting Bonferroni significant associations at p value < $4.93 \times 10^{-6}$. Source data are provided as a Source Data file.

with other common markers of inflammation and have been implicated in inflammatory diseases and cancer[7,8,27,28]. Genera from family *Ruminococcaceae* and *Lachnospiraceae* including genus *Blautia* are reported to be involved in conversion of primary bile acids into secondary bile acids and/or production of short-chain fatty acids (SCFAs)[25,29,30].

**Association of microbial diversity with circulating metabolites.** We next determined whether microbial diversity of gut microbiota was associated with lipoprotein particles or other metabolites (Fig. 2). When adjusting for multiple testing and age, sex, BMI, and medication use, the pattern emerging was that higher microbiome diversity was significantly associated with lower levels of VLDL particles (small, large, medium, very large, and extra-large), serum triglycerides, TotFA, MUFA, and SFA, and increased levels of large and extra-large HDL particles and an increased diameter of HDL (Fig. 2). As to the other metabolites, higher microbiome diversity was significantly associated with lower levels of glycoprotein acetyl, alanine, isoleucine, and lactate (Fig. 2). The strongest association was observed with triglycerides in VLDL particles (p value = $8.48 \times 10^{-10}$).

**Discussion**
We have examined the impact of gut microbiota on host circulating metabolites in 2309 individuals from the Rotterdam Study and LifeLines-DEEP cohort using $^1$H-NMR technology. We identified associations between the gut microbiota composition and various metabolites, including specific VLDL and HDL lipoprotein subfractions; serum lipid measures, including triglycerides and fatty acids; glycolysis-related metabolites, including lactate; ketone bodies, including acetate; amino acids, including alanine and isoleucine; and acute-phase reaction marker, including the glycoprotein acetyl independent on age, sex, BMI, and medication use. No associations were found to LDL subfractions except for triglycerides in small LDL and glucose levels measured by $^1$H-NMR.

Our results based on two large population-based studies identified associations between the gut microbiota composition and various lipoprotein particles. We observed an inverse association of family *Christensenellaceae* with VLDL particles of various sizes, small HDL particles, and triglycerides in medium HDL (Fig. 1b). The family *Christensenellaceae* was previously linked to BMI and was associated with the reduced weight gain as reported in the mice study in which germfree mice were

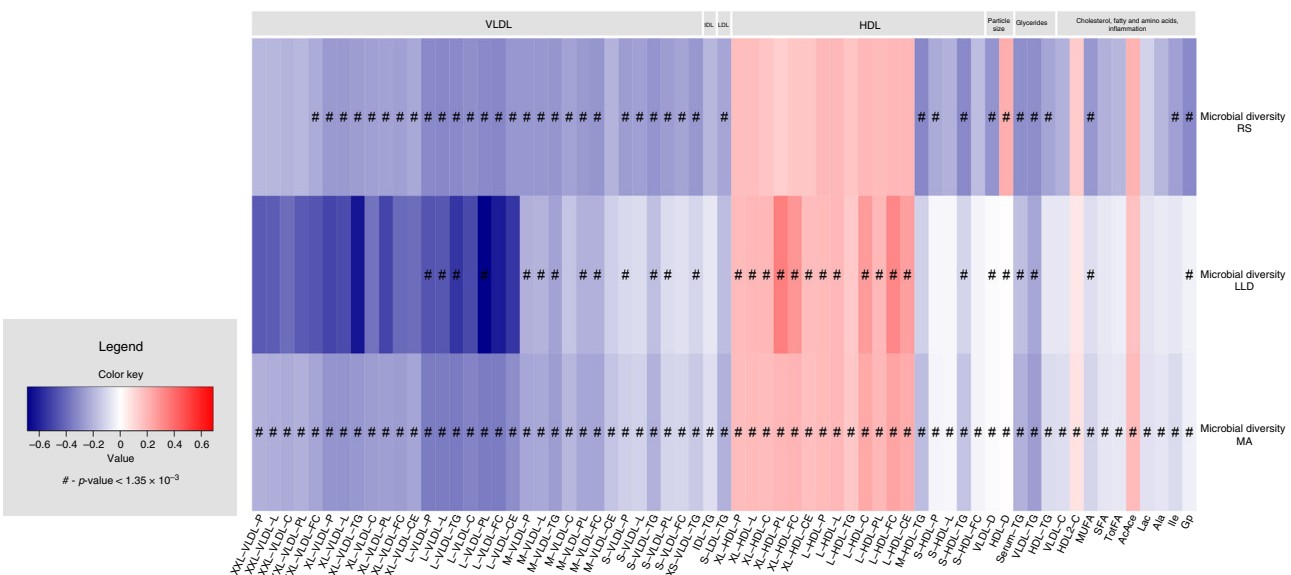

**Fig. 2 Results of association analysis between metabolites and alpha diversity (n = 2309).** Metabolites are displayed on x-axis whereas microbial diversity in the RS, LLD, and combined meta-analysis is shown on y-axis. Lipoprotein classes, include very-low-density lipoprotein particles (VLDL), low-density lipoprotein particles (LDL), and high-density lipoprotein particles (HDL) of very low (XS), low (S), medium (M), large (L), very large (XL), and extremely large (XXL) size. The colors represent effect estimates. Blue color stands for inverse association. Red color denotes positive associations. Symbols on the plot represent level of significance with hash denoting Bonferroni significant associations.

inoculated with lean and obese human fecal samples[20]. Furthermore, the family *Christensenellaceae* was reported to be the most heritable microbial taxon in the study by Goodrich et al. independently of the effect of BMI [20].

Interestingly, the gut microbiota composition showed association with VLDL and HDL particles of various sizes, however, weak association has been found for LDL and IDL particles suggesting that gut microbiota affects distinct classes of lipoproteins[31]. While VLDL particles of various sizes showed the same pattern of association, differences were noticed between large, medium, and small HDL particles suggesting that they are heterogeneous structures[32]. Small HDL particles are dense, protein-rich, and lipid-poor, whereas large HDL particles are large, lipid-rich particles[33,34]. Despite the fact that HDL is consistently associated with a reduced risk of cardiovascular disease, the past decade has seen major controversies on the clinical relevance of HDL interventions. Most trials aiming to increase HDL levels in the aggregate have been unsuccessful and were even stopped because of adverse effects[35,36]. The heterogeneity of HDL classes has been long recognized but can now be assessed on a large scale. This compositional heterogeneity of HDL results in functional heterogeneity such that small and large HDL particles are negatively correlated and display inverse relationship with various diseases, including cardiovascular disease, as reported previously[32,33]. As observed in our study the small HDL particles were associated with genus *Blautia* and family *Lachnospiraceae* and with lower diversity. Indeed the higher levels of small lipoprotein particle concentration have previously been associated with increased risk of stroke as reported in a recently published study of Holmes et al., while the large and extra-large HDL particles that were associated with family *Clostridiacea1*, genus *Clostridium sensu stricto1*, and unknown family and genus, were associated with decreased risk of cardiovascular disease and stroke[7]. Interestingly, family *Clostridiacea1* was previously inversely correlated with BMI, serum triglycerides and is known to be involved in bile acid metabolism [4,22].

Furthermore, we confirmed the association of genus *Ruminococcus gnavus* group and serum triglycerides level[37]. *Ruminococcus*

*gnavus* group was previously associated with low gut microbial richness[23] and its abundance was higher in patients with atherosclerotic cardiovascular disease[38]. This genus showed positive correlation with family *Lachnospiraceae* and inverse correlation with genera from family *Ruminococcaceae*. This is in line with earlier misclassification as a *Ruminococceae* (now a *Lachnospiraceae*)[39]. Microbial taxa that showed association with serum triglycerides showed association with other lipoprotein particles as well (Supplementary Fig. 13). However, we also observed microbial taxa that were exclusively associated with VLDL (two microbial genera), HDL (nine microbial genera and families) and LDL particles (one genus) suggesting that lipoprotein particles are important and not just a spillover effect of circulating triglycerides.

In addition to circulating lipids and lipoprotein particles, an association was found between gut microbiota and ketone bodies including acetate, amino acids including isoleucine, and acute-phase reaction marker, including glycoprotein acetyls mainly alpha 1. Circulating levels of acetate were specifically associated with genus *Ruminococcaceae UCG-005*. Acetate is the most common SCFA formed by bacterial species in the colon[40]. SCFA can serve as an energy source, predominantly via metabolism in liver[41,42]. Previous studies suggested that acetate mediates a microbiota-brain axis and promotes metabolic syndrome[43]. Circulating levels of isoleucine, an essential branched-chain amino acid, were inversely associated with three microbial taxa in our sample. Recent studies reported association of circulating levels of isoleucine with diabetes and cardiovascular disease[8,44]. Furthermore, isoleucine was reported to be negatively associated with *Christensenellaceae* and microbial diversity and positively with *Blautia*[45]. Even though we observe the same pattern of association between isoleucine and these taxa, the associations did not reach the significance threshold. Also recently, a study focusing on relation of fecal metabolites using mass spectroscopy (Metabolon) and the gut microbiota was published[6]. Even though the overlap of measured metabolites is limited, amino acids are measured on both platforms. Other amino acids showed a strong association with the gut microbiota but not isoleucine[6]. However, the concentration of metabolite levels in feces and blood may

differ. This is an important field of future research. Lastly, glycoprotein acetyls, a composite marker that integrates protein levels and glycosylation states of the most abundant acute phase proteins in circulation[46,47], was positively associated with genus *Blautia* and *Ruminococcus gnavus* group and negatively associated with microbial diversity. Genus *Blautia* is one of the microbial taxa with substantial heritability in twin study[20], and showed strong association with the host genetic determinants which has been associated with BMI and obesity[26]. *Blautia* was also reported to be involved in conversion of primary bile acids into secondary bile acids[24–26]. Glycoprotein acetyls are associated with other common markers of inflammation[46,47]. Circulating level of glycoprotein acetyls have been implicated in inflammatory diseases and cancer, and have been associated with mortality and cardiovascular disease [7,8,27,28].

Potential mechanisms through which gut microbiota may affect circulating lipid levels may involve bile acids and SCFAs. Some of the microbial taxa identified in our study are involved in conversion of primary to secondary bile acids and production of SCFAs. Previous studies demonstrated that hepatic and/or systemic lipid and glucose metabolism can be modulated by bacterially derived bile acids absorbed into bloodstream[31,48]. Another potential part of the biological basis for the association between circulating lipid levels and microbial taxa may be through SCFAs. SCFAs, such as butyrate, propionate, and acetate can affect lipid biosynthesis, serve as important energy source, and regulate inflammation and oxidative stress [49,50].

We also confirmed association of microbial diversity and serum triglycerides and provided insights into association with HDL particles[4]. Previous studies reported positive association between microbial diversity and HDL, however, advanced analysis on lipoprotein subfractions revealed that large and extra-large HDL particles were positivly associated with microbial diversity while negative association was found for medium and small HDL. Lower microbial diversity has been found in autoimmune diseases, obesity, and cardiometabolic conditions [51].

The strengths of our study are large sample size, population-based study design, and harmonized analysis in participating studies while correcting for factors such as use of medication and BMI. Combing the data of two large population-based studies allowed us to improve the statistical power of the study and internally cross-check consistency of the findings. However, our study has also limitations. When exploring circulating molecules, we focused on metabolites measured by Nightingale platform which covers a wide range of circulating compounds[52]. However, these compounds represent a limited proportion of circulating metabolites, therefore, future studies should focus on metabolites detected by other more detailed techniques[53]. Further, the gut microbial composition was determined from fecal samples. As gut microbial composition varies throughout the gut with respect to the anatomic location along the gut and at the given site, more complete picture of the gut microbiota could be obtained by getting samples from different locations along the intestines in the future[31,48]. Furthermore, when exploring gut microbiota, we focused on 16S rRNA sequencing. Even though broad shifts in community diversity could be captured by 16S rRNA, metagenomics approaches provide better resolution and sensitivity[54]. Additionally, the cross-sectional nature of our study failed to track changes within each individual. Future studies should focus on collecting stool and blood samples overtime for assessment of longitudinal changes. Finally, although our analyses were adjusted for various known confounders, residual confounding remains possible.

To conclude, we found association between gut microbiota composition and various circulating metabolites including lipoprotein subfractions, serum lipid measures, glycolysis-related metabolites, ketone bodies, amino acids, and acute-phase reaction markers. Association between gut microbiota and specific lipoprotein subfractions of VLDL and HDL particles provides insights into the role of microbiota in influencing host lipid levels. These observations support the potential of gut microbiota as a target for therapeutic and preventive interventions.

## Methods

**Study population.** Our study population included participants from two Dutch population cohorts: Rotterdam Study and LifeLines-DEEP.

The Rotterdam Study is a prospective population-based cohort study that includes participants from the well-defined district of Rotterdam[11]. The aim of Rotterdam Study is to investigate occurrence and determinants of diseases in elderly[11,55]. The initial cohort included 7983 persons aged 55 years or older in 1990 (RS-I)[11]. The cohort was further extended in 2000/2001 by additional 3011 individuals, aged 55 years and older (RS-II), and in 2006/2008 by adding 3932 individuals, aged 45 years and older (RS-III)[11]. The participants underwent interview at home and extensive set of examinations at the baseline[11]. Health status, anthropometric and clinical variables were assessed in a standardized manner by trained paramedical assistants and physicians in a specially built research facility in the center of the district[55]. These examinations were repeated every 3–4 years during the follow-up rounds in characteristics that could change over time[11]. All participants provided written informed consent. The institutional review board (Medical Ethics Committee) of the Erasmus Medical Center and by the review board of The Netherlands Ministry of Health, Welfare, and Sports approved the study.

The LifeLines-DEEP cohort is a sub-cohort of LifeLines study, a prospective population-based cohort study in the north of the Netherlands established[12]. The LifeLines cohort was established in 2006 among participants aged from 6 months to 93 years in order to gain insights into the etiology of healthy aging[56]. At the baseline, the participants filled in extensive questionnaires and visited one of the LifeLines Research Sites twice for physical examinations[56]. After completion of inclusion in 2013, the cohort includes 165,000 participants[12]. A follow-up questionnaire was sent to each participant every 18 months and follow-up measurements of the health parameters were performed every 5 years[56]. A subset of approximately 1500 LifeLines participants aged 18–81 years was included in Lifelines-DEEP[56]. These participants were examined more thoroughly, specifically with respect to molecular data. Additional biological materials and information on health status were collected for these participants[56]. The LifeLines-DEEP study is approved by the Ethical Committee of the University Medical Center Groningen[56]. All participants provided written informed consent.

**Metabolite profiling.** Quantification of small compounds in fasting plasma samples was performed using $^1$H-NMR technology in both participating studies[52,57,58]. Plasma samples of Rotterdam Study participants were collected in EDTA coated tubes during the visit to the research facility in the center of the district[11], while the plasma samples of LifeLines-DEEP participants were collected during participant's second visit to the site[56]. Simultaneous quantification of a wide range of metabolites, including amino acids, glycolysis-related metabolites, ketone bodies, fatty acids, routine lipids, and lipoprotein subclasses was done using the Nightingale Health metabolomics platform (Helsinki, Finland). Detailed description of the method can be found elsewhere[57,59]. In total there were 145 nonderived metabolite measures quantified in absolute concentration units across the participating studies (Supplementary Data 1).

**Gut microbiota profiling.** In order to study gut microbiota, fecal samples were collected from participants of Rotterdam Study and LifeLines-DEEP study. Fecal samples of Rotterdam Study participants were collected at home and sent through regular email to the Erasmus MC. Upon arrival at Erasmus MC, samples were stored at −20 °Csd an automated DNA isolation kit (Arrow; DiaSorin S.P.A., Saluggia, Italy) according to the manufacturer's protocol. The V3 and V4 variable regions of the 16S rRNA gene were amplified using the 319F (ACTCCTACGG-GAGGCAGCAG) −806 (RGGACTACHVGGGTWTCTAAT) primer pair and dual indexing and sequenced on Illumina MiSeq sequencer (MiSeq Reagent Kit v3, 2 × 300 bp)[56,60]. Fecal samples of Life-Lines-DEEP participants were collected at home and stored immediately at −20 °C. After transport on dry ice, all samples were stored at −80 °C. Aliquots were made, and DNA was isolated with the All-Prep DNA/RNA Mini Kit (Qiagen; cat. #80204). Isolated DNA was sequenced at the Broad Institute, Boston, using Illumina MiSeq paired-ends flanking. Hypervariable region V4 was selected using forward primer 515F (GTGCCAGCMGC CGCGGTAA) and reverse primer 806R (GGACTACHVGGGTWTCTAAT)[56]. A direct classification of 16S sequencing reads using RDP classifier (2.12) and SILVA 16S database (release 128) were used to reconstruct taxonomic composition of studied communities, with binning posterior probability cutoff of 0.8[61]. All 16S libraries were rarefed to 10,000 reads prior to taxonomy binning. This operational taxonomic unit-independent approach was utilized to decrease domain-dependent bias. The microbial Shannon diversity index was calculated on taxonomic level of

genera, using vegan package in R (https://www.r-project.org/). Gut microbiota composition dataset included 1427 participants from the RS-III cohort that participated in the second examination round at the study center. Metabolite measurements were available for 1390 Rotterdam Study (RSIII-2) participants. In the LifeLines-DEEP study, gut microbiota composition dataset, included 1186 participants; from them the metabolite measurements were available for 915 participants.

**Statistical analysis**. Prior to the analysis, all metabolites were natural logarithmic transformed to reduce skewness of traits distributions. To deal with values under detectable limit (reported as zeros) we added half of the minimum detectable value of the corresponding metabolite prior to transformation. The metabolite measures were then centered and scaled to mean of 0 and standard deviation of 1. Similarly, to reduce skewness of the distribution of microbial taxa counts, we first added 1 to all taxonomy counts and then performed natural log transformation. Correlation between microbial taxa was assessed by Spearman correlation.

The relationship between metabolites and microbial taxa was assessed by linear regression analysis while adjusting for age, sex, BMI, technical covariates including time in mail and DNA batch effect (only in Rotterdam Study) and regular medication use (yes/no) including lipid-lowering medication (395 users in Rotterdam Study and 34 in Lifelines-DEEP), proton-pump inhibitors (258 users in Rotterdam Study and 72 in Lifelines-DEEP), and metformin (67 users in Rotterdam Study and 8 in Lifelines-DEEP). The analyses were further adjusted for smoking status defined as never, former or current and daily alcohol consumption (in grams per day). Participants using antibiotics were excluded from the analysis. The summary statistics of participating studies were combined using inverse variance-weighted fixed-effect meta-analysis using rmeta package in R (https://cran.r-project.org/web/packages/rmeta/index.html, https://www.r-project.org/). In total, 145 overlapping metabolite measures and 345 overlapping microbial taxa at taxonomic level of family and genera were tested for association. These microbial taxa were present in at least three samples. As measurements in both metabolomics and gut microbiota datasets are highly correlated, we used a method of Li and Ji[14] to calculate a number of independent tests. There were 37 independent tests among the metabolite measures and 274 independent tests among microbial taxa. The significance threshold was thus set at $0.05/ (37 \times 274) = 4.93 \times 10^{-6}$.

The relationship between metabolites and microbial diversity was also assessed by linear regression analysis while adjusting for age, sex, BMI, technical covariates, and medication use (lipid-lowering medication, protein-pump inhibitors, and metformin) in each of the participating studies and summary statistics results were combined using inverse variance-weighted fixed-effect meta-analysis using rmeta package in R.

**Reporting summary**. Further information on research design is available in the Nature Research Reporting Summary linked to this article.

## Data availability

All relevant data supporting the key findings of this study are available within the article and its supplementary information files. Data underlying Fig. 1 and Supplementary Figs. 1–13 are provided as Source Data file. Other data are available from the corresponding author upon reasonable requests. Due to ethical and legal restrictions, individual-level data of the Rotterdam Study (RS) cannot be made publicly available. Data are available upon request to the data manager of the Rotterdam Study Frank van Rooij (f.vanrooij@erasmusmc.nl) and subject to local rules and regulations. This includes submitting a proposal to the management team of RS, where upon approval, analysis needs to be done on a local server with protected access, complying with GDPR regulations. The LifeLines-DEEP metagenomics sequencing data are available at the European Genome-phenome Archive under accession EGAS00001001704.

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

## Acknowledgements

*Rotterdam Study*: The Rotterdam Study is funded by Erasmus Medical Center and Erasmus University, Rotterdam, Netherlands Organization for the Health Research and Development (ZonMw), the Research Institute for Diseases in the Elderly (RIDE), the Ministry of Education, Culture and Science, the Ministry for Health, Welfare and Sports, the European Commission (DG XII), and the Municipality of Rotterdam. This work was performed within the framework of the Biobanking and BioMolecular resources Research Infrastructure (BBMRI) Metabolomics Consortium funded by BBMRI-NL, a research infrastructure financed by the Dutch government (Netherlands Organization for Scientific Research [NWO], nos. 184.021.007 and 184033111), the CardioVasculair Onderzoek Nederland (CVON 2012-03), the Common mechanisms and pathways in Stroke and Alzheimer's disease (CoSTREAM) project (www.costream.eu, grant agreement No. 667375), Memorabel program (project number 733050814) and U01-AG061359 NIA. Djawad Radjabzadeh was funded by an Erasmus MC mRACE grant "Profiling of the human gut microbiome". The generation and management of stool microbiome data for the Rotterdam Study (RSIII-2) were executed by the Human Genotyping Facility of the Genetic Laboratory of the Department of Internal Medicine, Erasmus MC, Rotterdam, The Netherlands. We thank Nahid El Faquir and Jolande Verkroost-Van Heemst for their help in sample collection and registration, and Pelle van der Wal, Kamal Arabe, Hedayat Razawy and Karan Singh Asra for their help in DNA isolation and sequencing. Furthermore, we thank drs. Jeroen Raes and Jun Wang (KU Leuven, Belgium) for their guidance in 16S rRNA profiling and dataset generation. The authors are grateful to the study participants, the staff from the Rotterdam Study and the participating general practitioners and pharmacists. *LifeLines DEEP*: LifeLines-DEEP project was funded by the Netherlands Heart Foundation (IN-CONTROL CVON grant 2012-03 to A.Z. and J.F.); by Top Institute Food and Nutrition, Wageningen, The Netherlands (TiFN GH001 to C.W.); by the Netherlands Organization for Scientific Research (NWO) (NWO-VIDI 864.13.013 to J.F., NWO-VIDI 016.178.056 to A.Z., NWO-VIDI 917.14.374 to L.F., NWO Spinoza Prize SPI 92-266 to C.W., and NWO Gravitation Netherlands Organ-on-Chip Initiative (024.003.001) to C.W.); by the European Research Council (ERC) (FP7/2007-2013/ERC Advanced Grant Agreement 2012-322698 to C.W., ERC Starting Grant 715772 to A.Z., and ERC Starting Grant 637640 to L.F.); by the Stiftelsen Kristian Gerhard Jebsen Foundation (Norway) to C.W.; and by the RuG Investment Agenda Grant Personalized Health to C.W. A.Z. also holds a Rosalind Franklin Fellowship from the University of Groningen. The authors thank participants and staff of the LifeLines-DEEP cohort for their collaboration. We thank J. Dekens, M. Platteel, A. Maatman, and J. Arends for management and technical support.

## Author contributions

Conception and design of the study: D.V., D.R., A.K., J.F., R.K., C.M.v.D. Collection of the data: D.R., R.K., A.K., A.Z., M.A.I, A.G.U., L.F., C.W. and J.F. Analysis: D.V., D.R. and A.K. Interpretation of the data: D.V., D.R., A.K., A.Z., M.A.I, A.G.U., N.A., R.K., J.F., C.M.v.D. Drafting of the paper: D.V., C.M.v.D. All authors read, revised, and approved the final draft.

## Competing interests

The authors declare no competing interests.
