## [Peer Review File · Nature Communications]

Reviewers' comments:

Reviewer #1 (Remarks to the Author):

This is a timely, interesting, reasonably novel and well-written contribution. I generally like how the results are presented and discussed. However, in some places I would suggest toning down the causality interpretation (though the direction from microbiota to systemic metabolism is likely, there are multiple confounding factors and the causal pathways may vary for different metabolic measures) eg in the discussion “driven by” would be better replaced by “association language”.

I don't have any detailed criticism but I think the authors have done good work and the paper is easy to read and understand. However, I find the analyses (and discussion) somewhat limited and would find it a valuable addition to show the correlation structure in the microbial genera and families (a correlation heat map). In addition, and in relation to some contributions already existing in the literature and also cited in this work, it would be valuable to analyse the correlation network for the microbial and metabolic measures together and discuss these findings in more detail regarding the literature.

Would hierarchical clustering add anything to the interpretation of the current results? Maybe would allow a little bit more focus on the key role of circulating triglycerides? I would find that useful. How important the individual lipoprotein fractions are or is it just a spillover effect of high circulating triglycerides? Please comment in the discussion.

Would it be better to show just B in Figure 1 in the actual paper and move A to a supplement? Then the microbial genera and families would be easier to see. Hierarchical clustering in this dimension maybe?

The authors should also make the comment on data availability more accurate; what are the conditions for the data access and what data would be made available. Would it be possible for the authors to make a freely accessible site for the key data in the study?

Reviewer #2 (Remarks to the Author):

In this original investigation by Vojinovic et al, two large population-based cohorts were combined to associate gut microbial features with circulating metabolites and biomarkers. They found 32 microbial traits were significantly associated with various metabolites, cardiometabolic biomarkers, and amino acids.

Although the manuscript is generally well done, there are several limitations and areas which could be improved. The manuscript is quite brief and would benefit from greater critical evaluation and discussion of what truly makes their study and its findings unique, novel, and a clear step forward. While the authors ultimately conclude that the associations found “provide novel insights into the role of microbiota in influencing host lipid levels,” too little text is devoted to supporting this claim with most results simply listed as a series of tests that reach statistical significance, with scant information on biological interpretation. In its present form, the manuscript will require an extensive expansion of both text and analyses.

Results: This section could benefit from a bit more (but brief) study overview/study population description for readers navigating the manuscript from Intro to Results (saving Methods for the end). While clear to some, it may also be helpful to explicitly state why certain factors were chosen for multivariable adjustment. Additionally, the Results section currently reads as a long list of positive association tests with very little in the way of the implications of these findings, particularly pages 5-6. Little context is offered to help readers determine if a given metabolite or microbe is considered harmful or beneficial to human health, which would present a significant challenge to this journal’s wide and diverse readership.

Page 5, paragraph 1: Greater detail regarding which medications and how multiple testing was dealt with should be provided, even if elaborated upon in the Methods section.

Page 6, paragraph 1: “In addition, there were more targeted associations..” may be interpreted as there were associations among a more targeted set of tests, but I believe the authors are trying to convey that they were able to re-demonstrate several known associations?

Page 6, paragraph 3: There is a reference to “Supplementary 3,” believed to be Sup. Table 3.

Discussion: Page 10, paragraph 2: This does not appear to be a correct statement: “Merging the data of two large population-based studies allowed us to internally validate the findings”. Not clear what is meant by this since the analyses are really done as combination of the two cohorts.

Methods: The section on Study Population could be greatly expanded to include additional cohort details, including how they are clinically phenotyped, how and when plasma and stool were obtained, whether the cohorts were conceived for any particular reason, what follow up constitutes active participation, and more.

I do not think that the multivariable model chosen should include technical covariates if it is only recorded in one cohort. Greater details should be offered with respect to the covariates, i.e. are medications yes/no, regular use or at time of sampling?, is smoking in pack-years, alcohol in grams? Were these variables discretized?

Of the 455 overlapping taxa tested, were these subject to any prevalence or abundance filtering?

Figure 1: Overall, this figure could be greatly improved in several ways. 1) an in-figure legend depicting the range of blue to red i.e. association values, as well as the use of a different symbol (the # sign obscures much of the cell). 2) A facet break between metabolite groups tested (i.e. AAs from cholesterols, etc) may also help with interpretability. The test for association should also be detailed in the caption, along with the significance level depicted with the symbol. There is a clear block structure to the heatmap, suggesting strong collinearity in the metabolites. This could be of biological importance if the authors can make a better case as to why different sized metabolites matter (and are worthy of being separately studied in this manner).

Figure 2 could be made more clear. It would benefit from a figure legend and more information on test of significance and range of values. Also, the use of “beta” in the columns may lead to confusion. Very little text in the manuscript is devoted to Figure 2.

Supplementary Table 2 and 3: In the text, these data are used to support the notion that, “The direction of effect size across the cohorts was generally concordant” but this can be challenging to see across columns, and may be better represented by bar/whisker graphs with betas/SEs by cohort and overall. It also appears that only statistically significant findings are detailed. While Supplementary Table 1 offers the list of metabolites tested, if the prior statement is correct, there is nowhere in the text or supplementary material detailing the exact microbial feature set tested. This is especially noteworthy given testing demonstrates that analyses appear to combine microbial traits of different taxonomic ranks, i.e. family with genus, which could just be due to understandable inability to resolve OTUs, or worse, the erroneous inclusion of all within the same testing set.

The data set should be made available as per Nature Communications guidelines.

Response to Reviewers: NCOMMS-19-09554 (Vojinovic et al.)

We would like to express our thanks to the reviewers for their constructive comments and criticisms. Our specific responses to reviewer concerns are described below (reviewer comments are in bold; our responses are in normal text).

Reviewer #1 (Remarks to the Author):

This is a timely, interesting, reasonably novel and well-written contribution. I generally like how the results are presented and discussed. However, in some places I would suggest toning down the causality interpretation (though the direction from microbiota to systemic metabolism is likely, there are multiple confounding factors and the causal pathways may vary for different metabolic measures) eg in the discussion “driven by” would be better replaced by “association language”.

Complying with the reviewer’s suggestion, we have toned down the causality interpretation in the relevant sections of discussion in the revised version of the manuscript on page 9 (line 176) and page 10 (line 179). We have replaced “driven by” with “associated with”.

I don’t have any detailed criticism but I think the authors have done good work and the paper is easy to read and understand. However, I find the analyses (and discussion) somewhat limited and would find it a valuable addition to show the correlation structure in the microbial genera and families (a correlation heat map). In addition, and in relation to some contributions already existing in the literature and also cited in this work, it would be valuable to analyse the correlation network for the microbial and metabolic measures together and discuss these findings in more detail regarding the literature.

We have now provided the correlation structure in microbial genera and families and metabolic measures together in the revised version of the manuscript. The correlation structure is illustrated in the figure below. As the inclusion of all microbial taxa and metabolites in one heat map plot affects the readability of text along the axis, we have also presented these results in Supplementary Table 5. When analyzing microbial and metabolic measures together, we observed a weak correlation between the two. Spearman correlation coefficient for majority of microbial taxa and metabolites was around 0. On the other hand, we noticed that microbial taxa within the same families as well as some families tend to cluster together. For example, the correlation between family *Lachnospiraceae* and genus *Blautia* that belongs to this family was 0.82 (page 7, lines 116-118). These microbial taxa were associated with small HDL particles. Further, several microbial taxa were identified to be associated with VLDL particles of various sizes in our study. Correlation analysis between these microbial taxa revealed a positive correlation between family *Christensenellaceae*, *Christensenellaceae* R7 group genus and *Ruminococcaceae* genera (ρ ranged between 0.50 and 0.67) as well as within genera from *Ruminococcaceae* family (ρ ranged between 0.46 and 0.77) (page 6, lines 93-97). These correlations are not unexpected and were reported by previous studies as well.¹² Additionally, we identified association of *Ruminococcus gnavus* group and triglycerides

¹ Oki, Kaihei, et al. "Comprehensive analysis of the fecal microbiota of healthy Japanese adults reveals a new bacterial lineage associated with a phenotype characterized by a high frequency of bowel movements and a lean body type." *BMC microbiology* 16.1 (2016): 284.

² Ayeni, Funmilola A., et al. "Infant and adult gut microbiome and metabolome in rural Bassa and urban settlers from Nigeria." *Cell reports* 23.10 (2018): 3056-3067.

and glycoprotein acetyls. This genus showed a positive correlation with family *Lachnospiraceae* and an inverse correlation with genera from family *Ruminococcaceae*. This is in line with earlier misclassification of this genus as a *Ruminococceae* (now a *Lachnospiraceae*) (page 10, lines 187-189).

Finally, the strongest correlation and clear blocks were observed between metabolic measurements. This is not surprising given the fact that majority of metabolic measurements analyzed in our study are lipoprotein subfractions which are known to be highly correlated.

Figure. Heat map plot showing correlation structure between microbial taxa and metabolites tested in our study. Red color stands for positive Spearman correlation coefficient while blue color denotes negative correlation. As the big number of observation included in heat map plot affects the readability of text along the axis, these results are also presented in the Supplementary Table 5.

genus) suggesting that individual lipoprotein particles are important and not just a spillover effect of high circulating triglycerides. We have clarified this in the relevant section of results and discussion in the revised version of the manuscript (page 7: lines 109-111, page 10, lines 189-194).

Figure. Venn diagram showing number of microbial taxa associated with circulating triglycerides, very-low, low and high-density lipoprotein particles.

Importance of high resolution lipoprotein subclass profiling has been highlighted in multiple studies as it improve our understanding of several diseases. It has been reported that abnormalities in low and very-low-density lipoprotein distribution are associated with metabolic disease etiology, cardiovascular diseases and type 2 diabetes.^{3,4,5,6,7,8,9} Also, multiple studies have highlighted that the inverse association of cholesterol to cardiovascular disease risk is primarily from lipids in large high-density lipoprotein particles, whereas lipids in small high-density lipoprotein particles have commonly not been associated with disease risk. We have clarified this on page 5, lines 83-86.

Would it be better to show just B in Figure 1 in the actual paper and move A to a supplement? Then the microbial genera and families would be easier to see. Hierarchical clustering in this dimension maybe?

Complying with the reviewer's suggestion, we have improved Figure 1 in such a way that the microbial genera and families are easier to see and read in the revised version of the manuscript. We have included hierarchical clustering of the microbial families and genera in Figure 1A. In order to be able to compare association pattern after further adjustment for alcohol consumption and smoking, the order of the

³ Pirillo, Angela, Giuseppe Danilo Norata, and Alberico Luigi Catapano. "High-density lipoprotein subfractions-what the clinicians need to know." *Cardiology* 124.2 (2013): 116-125.

⁴ Swanson, Barbara, et al. "Lipoprotein particle profiles by nuclear magnetic resonance spectroscopy in medically underserved HIV-infected persons." *Journal of clinical lipidology* 3.6 (2009): 379-384.

⁵ Lamarche, Benoit, et al. "Small, dense low-density lipoprotein particles as a predictor of the risk of ischemic heart disease in men: prospective results from the Qué bec Cardiovascular Study." *Circulation* 95.1 (1997): 69-75.

⁶ Krauss, Ronald M., and Darlene M. Dreon. "Low-density-lipoprotein subclasses and response to a low-fat diet in healthy men." *The American journal of clinical nutrition* 62.2 (1995): 478S-487S.

⁷ Krauss, Ronald M. "Lipids and lipoproteins in patients with type 2 diabetes." *Diabetes care* 27.6 (2004): 1496-1504.

⁸ Wang, J., et al. "Lipoprotein subclass profiles in individuals with varying degrees of glucose tolerance: a population-based study of 9399 Finnish men." *Journal of internal medicine* 272.6 (2012): 562-572.

⁹ Holmes, Michael V., et al. "Lipids, lipoproteins, and metabolites and risk of myocardial infarction and stroke." *Journal of the American College of Cardiology* 71.6 (2018): 620-632.

microbial families and genera presented in Figure 1B was kept the same as in Figure 1A. Additionally, as suggested by the reviewer, we have incorporated hierarchical clustering of the microbial families and genera in Figure 1B. Please see the Figure below. Clustering in this dimension does not change the Figure. If the reviewer wants we can put those results in the Supplementary Material of the manuscript.

Figure. Results of association analysis between metabolites and microbial genera and families assessed by linear regression analysis after adjustment for age, sex, body-mass index, technical covariates, medication use, smoking, and alcohol consumption.

The authors should also make the comment on data availability more accurate; what are the conditions for the data access and what data would be made available. Would it be possible for the authors to make a freely accessible site for the key data in the study?

We have now made data availability statement more accurate in the relevant section of the revised version of the manuscript (page 17, lines 356-360). We have clarified that the summary statistics are available in Supplementary Tables 3 and 4. The Rotterdam Study data could be accessed through consultation with the management team of the cohort. Due to the new General Data Protection Regulation (GDPR), we are no longer allowed to share pseudonymized data in open or closed data repositories. The LifeLines-DEEP metagenomics sequencing data are available at the European Genome-phenome Archive under accession EGAS00001001704.

Reviewer #2 (Remarks to the Author):

In this original investigation by Vojinovic et al, two large population-based cohorts were combined to associate gut microbial features with circulating metabolites and biomarkers. They found 32 microbial traits were significantly associated with various metabolites, cardiometabolic biomarkers, and amino acids.

Although the manuscript is generally well done, there are several limitations and areas which could be improved. The manuscript is quite brief and would benefit from greater critical evaluation and discussion of what truly makes their study and its findings unique, novel, and a clear step forward. While the authors ultimately conclude that the associations found “provide novel insights into the

role of microbiota in influencing host lipid levels,” too little text is devoted to supporting this claim with most results simply listed as a series of tests that reach statistical significance, with scant information on biological interpretation. In its present form, the manuscript will require an extensive expansion of both text and analyses.

Complying with the reviewer’s suggestion, we have now added text in the relevant section of results and discussion with information on biological interpretation of the findings. We have provided more context on given metabolite and/or microbial taxa in relation to human health/disease and potential mechanism through which gut microbiota may affect circulating metabolites.

With regard to circulating metabolites, previous studies highlighted the relation of VLDL lipoprotein particles of various sizes with metabolic disease etiology, cardiovascular diseases and type 2 diabetes, association of diameter of HDL particles with cardiovascular disease and glycoprotein acetyls with inflammatory diseases and cancer. We have clarified this on page 5: lines 83-86, page 7: lines 121-122 and 129-132 of the revised version of the manuscript. These as well as other circulating metabolites investigated in our study showed association with microbial genera and families including family *Christensenellaceae*, family *Clostridiaceae1*, genus *Ruminococcus gnavus* group, genus *Blautia* etc. Interestingly, these microbial taxa were previously linked to cardiovascular and metabolic conditions (page 6, lines 98-100). Among the microbial taxa that showed significant evidence of association with circulating metabolites were the genera from family *Lachnospiraceae* and *Ruminococcaceae* as well as genus *Blautia* which were reported to be involved in conversion of primary bile acids into secondary bile acids and/or production of short-chain fatty acids (SCFAs). Bacterially derived bile acids that are absorbed into the bloodstream and can modulate hepatic and/or systemic lipid and glucose, and SCFAs, such as acetate, propionate, and butyrate, are also absorbed from the gut and can have potent effects on energy expenditure and insulin sensitivity in peripheral metabolic tissues.^{10,11} Therefore, at least part of the biological basis for the association between microbial taxa and circulating lipoprotein levels may be through bile acids and SCFAs. We have clarified this on page 12, lines 223-230.

Results: This section could benefit from a bit more (but brief) study overview/study population description for readers navigating the manuscript from Intro to Results (saving Methods for the end). While clear to some, it may also be helpful to explicitly state why certain factors were chosen for multivariable adjustment. Additionally, the Results section currently reads as a long list of positive association tests with very little in the way of the implications of these findings, particularly pages 5-6. Little context is offered to help readers determine if a given metabolite or microbe is considered harmful or beneficial to human health, which would present a significant challenge to this journal’s wide and diverse readership.

Complying with the reviewer’s suggestion, we have now added a brief overview of the study population in the relevant part of results section in the revised version of the manuscript (page 4, lines 60-64) and have also clarified how and why certain factors were chosen for multivariable adjustment (page 5, lines

¹⁰ Ghazalpour, Anatole, et al. "Expanding role of gut microbiota in lipid metabolism." *Current opinion in lipidology* 27.2 (2016): 141.

¹¹ Chambers, Edward S., et al. "Role of gut Microbiota-Generated short-chain fatty acids in metabolic and cardiovascular health." *Current nutrition reports* 7.4 (2018): 198-206.

72-73). The variables corrected for in the analysis were selected according to previous literature reports focusing on factors related to gut microbiota composition and circulating metabolites.^{12,13}

With regard to the implications of the findings, we have also provided more context if a given metabolite or microbiome is considered harmful or beneficial for human health. For example, we have now clarified that family *Christensenellaceae* was previously associated with BMI, genus *Marvinbryantia* was linked to bowel dysfunction while family *Clostridiaceae1* is involved in bile acid metabolism and linked to BMI. These microbial taxa showed association with VLDL and HDL particles of various sizes. Previous studies highlighted association of abnormalities in VLDL distribution with metabolic disease etiology, cardiovascular diseases and type 2 diabetes^{14,15,16,17,18,19,20} and inverse association of very large and large HDL particles and small and medium HDL particles in relation to disease risk.¹⁶ Furthermore, we have clarified that diameter of HDL particles was linked to cardiovascular disease, while circulating levels of glycoprotein acetyls were linked to inflammatory diseases and cancer. Several genera from family *Ruminococcaceae* and genus *Blautia* were associated with glycoprotein levels in our study. These genera have been reported to be involved in conversion of primary bile acids into secondary bile acids. The changes have been made in the relevant section of the revised version of the manuscript (page 5: lines 83-86, page 6: lines 98-100, 104-105, page 7: lines 113-116, 121-122, 129-132).

Page 5, paragraph 1: Greater detail regarding which medications and how multiple testing was dealt with should be provided, even if elaborated upon in the Methods section.

Complying with the reviewer's suggestion, we have now added details regarding medications we adjusted for and how we dealt with multiple testing in the relevant section of the revised version of the manuscript. Please see page 5, lines 70-76.

Page 6, paragraph 1: "In addition, there were more targeted associations.." may be interpreted as there were associations among a more targeted set of tests, but I believe the authors are trying to convey that they were able to re-demonstrate several known associations?

We apologize for the confusion and have now rephrased the sentence in the revised version of the manuscript on page 6, lines 104-105. We have clarified that we were able to confirm some of the previously reported associations.

¹² Zhernakova, Alexandra, et al. "Population-based metagenomics analysis reveals markers for gut microbiome composition and diversity." *Science* 352.6285 (2016): 565-569.

¹³ Dunn, Warwick B., et al. "Molecular phenotyping of a UK population: defining the human serum metabolome." *Metabolomics* 11.1 (2015): 9-26.

¹⁴ Pirillo, Angela, Giuseppe Danilo Norata, and Alberico Luigi Catapano. "High-density lipoprotein subfractions-what the clinicians need to know." *Cardiology* 124.2 (2013): 116-125.

¹⁵ Swanson, Barbara, et al. "Lipoprotein particle profiles by nuclear magnetic resonance spectroscopy in medically underserved HIV-infected persons." *Journal of clinical lipidology* 3.6 (2009): 379-384.

¹⁶ Lamarche, Benoit, et al. "Small, dense low-density lipoprotein particles as a predictor of the risk of ischemic heart disease in men: prospective results from the Québec Cardiovascular Study." *Circulation* 95.1 (1997): 69-75.

¹⁷ Krauss, Ronald M., and Darlene M. Dreon. "Low-density-lipoprotein subclasses and response to a low-fat diet in healthy men." *The American journal of clinical nutrition* 62.2 (1995): 478S-487S.

¹⁸ Krauss, Ronald M. "Lipids and lipoproteins in patients with type 2 diabetes." *Diabetes care* 27.6 (2004): 1496-1504.

¹⁹ Wang, J., et al. "Lipoprotein subclass profiles in individuals with varying degrees of glucose tolerance: a population-based study of 9399 Finnish men." *Journal of internal medicine* 272.6 (2012): 562-572.

²⁰ Holmes, Michael V., et al. "Lipids, lipoproteins, and metabolites and risk of myocardial infarction and stroke." *Journal of the American College of Cardiology* 71.6 (2018): 620-632.

Page 6, paragraph 3: There is a reference to “Supplementary 3,” believed to be Sup. Table 3.

We thank the reviewer for pointing this out. We have replaced Supplementary with Supplementary Table on page 6.

Discussion: Page 10, paragraph 2: This does not appear to be a correct statement: “Merging the data of two large population-based studies allowed us to internally validate the findings”. Not clear what is meant by this since the analyses are really done as a combination of the two cohorts.

We have now rephrased the sentence stating that the analyses were done as combination of two large population-based studies in the relevant section of the revised version of the manuscript (page 12, lines 239-240).

Methods: The section on Study Population could be greatly expanded to include additional cohort details, including how they are clinically phenotyped, how and when plasma and stool were obtained, whether the cohorts were conceived for any particular reason, what follow up constitutes active participation, and more.

Complying with the reviewer’s suggestion, we have now provided additional cohort details in the revised version of the manuscript (pages 13, lines 266-275 and lines 280-288). We have clarified that the Rotterdam Study was designed in order to investigate occurrence and determinants of diseases in the elderly. The initial cohort included 7,983 persons living in the well-defined district in Rotterdam who were aged 55 years or older in 1990 (RS-I). The cohort was further extended in 2000/2001 by additional 3,011 individuals, aged 55 years and older (RS-II), and in 2006/2008 by adding 3,932 individuals, aged 45 years and older (RS-III). The participants underwent interview at home and extensive set of examinations at the baseline. Health status, putative risk factors, and anthropometric and clinical variables were assessed in a standardized manner by trained paramedical assistants and physicians in a specially built research facility in the center of the district. These examinations were repeated every 3-4 years during the follow-up rounds in characteristic that could change over time. We have also clarified that LifeLines cohort is established in order to gain insights into the etiology of healthy aging. The participants filled in extensive questionnaires and visited one of the LifeLines Research Sites for physical examinations at the baseline. After completion of inclusion in 2013, the cohort includes 165,000 participants. A follow-up questionnaire was sent to each participant every 18 months and follow-up measurements of the health parameters were performed every 5 years. A subset of approximately 1,500 LifeLines participants aged 18-81 years was included in Lifelines-DEEP. These participants were examined more thoroughly, specifically with respect to molecular data. For these participants, additional biological materials and information on health status were collected.

We have also clarified that plasma samples of Rotterdam Study participants which were collected in EDTA coated tubes during the visit to the research facility in the center of the district were used to quantify metabolites. Further, the plasma samples of LifeLines-DEEP participants were collected during the second visit to the cite. Please see page 15, lines 293-296. We have also provided details regarding the collection of stool samples. Fecal samples of Rotterdam Study participants were collected at home and sent through regular email to Erasmus MC where they were stored at -20°C, whereas the fecal samples of life-Lines-DEEP participants were collected at home and stored immediately at -20°C. Please see the relevant section of the revised version of the manuscript on page 15, lines 304-318.

I do not think that the multivariable model chosen should include technical covariates if it is only recorded in one cohort. Greater details should be offered with respect to the covariates, i.e. are medications yes/no, regular use or at time of sampling?, is smoking in pack-years, alcohol in grams? Were these variables discretized?

We have rerun the analysis in the Rotterdam Study using the model with all covariates except DNA batch effect. We have further compared these results with the results generated using the model with all covariates including DNA batch effect. The results are illustrated in the plot below. Effect estimates using the model with all covariates except DNA batch effect are displayed on horizontal axis while effect estimates when using the model with all covariates including DNA batch effect are displayed on vertical axis. Even though the reviewer is right, the high correlation of effect estimates in two models (cor estimate = 0.97) suggests that the results did not change.

Figure. Plot comparing effect estimates in the Rotterdam Study using the linear regression model with and without DNA batch effect.

We have also provided greater details regarding the covariates used in our analysis in the revised version of the manuscript (page 17, lines 335-336 and 339-340). We have clarified that the regular medication use was defined as yes/no, smoking status as never, former and current and alcohol consumption as daily consumption in grams.

Of the 455 overlapping taxa tested, were these subject to any prevalence or abundance filtering?

We have now clarified in the relevant section of the revised version of the manuscript that we focused on the taxa present in at least 3 samples (page 17, lines 345-346).

Figure 1: Overall, this figure could be greatly improved in several ways. 1) an in-figure legend depicting the range of blue to red i.e. association values, as well as the use of a different symbol (the # sign obscures much of the cell). 2) A facet break between metabolite groups tested (i.e. AAs from cholesterol, etc) may also help with interpretability. The test for association should also be detailed in the caption, along with the significance level depicted with the symbol. There is a clear block

structure to the heatmap, suggesting strong collinearity in the metabolites. This could be of biological importance if the authors can make a better case as to why different sized metabolites matter (and are worthy of being separately studied in this manner).

Complying with the reviewer's suggestion we have included the legend depicting the range of blue to red association values and explanation of the hash symbol in the revised version of Figure 1. We have also clarified metabolite groups that were tested. Further, we have provided additional details in the caption for Figure 1 including test for association and significance level depicted with the hash symbol (page 27).

With regard to the reviewer's question why different size metabolites matter, we have now provided evidence from previous publications investigating these metabolites in relation to disease. In a recently published study of Holmes et al. metabolites of different sizes were studied in relation to stroke and myocardial infarction (MI). The authors found the majority of VLDL particle subclasses were positively associated with risk of ischemic stroke; however, the associations were somewhat weaker for very small VLDL and IDL as compared with MI. Furthermore, the authors describe other metabolites such as particle size of small VLDL, cholesterol in large HDL particles, mean diameter of HDL, etc. to be associated with myocardial infarction but not stroke. These findings suggest that different sized metabolites are worthy of being separately studied (page 9, lines 173-181).

Figure 2 could be made more clear. It would benefit from a figure legend and more information on test of significance and range of values. Also, the use of "beta" in the columns may lead to confusion. Very little text in the manuscript is devoted to Figure 2.

We have now included the figure legend with information on a range of values and significant threshold in the revised version of the manuscript. We have also rephrased the row names. Instead of using term "beta", we have clarified that we evaluated the relation between microbial diversity and circulating metabolites. To help with the interpretability of the results, we have also clarified metabolite groups that were tested.

With regard to the text devoted to Figure 2, we have now included a paragraph on association of microbial diversity and circulating metabolites in the relevant section of the revised version of the manuscript. We have now discussed that association of microbial diversity and serum triglycerides was previously reported as well as association with HDL. However, advanced analysis on lipoprotein substructures provided novel insights into association with HDL particles as the positive association was found for large and extra-large particles while negative association was observed for medium and small HDL particles. Furthermore, we discussed that higher microbiome diversity was associated with lower levels of glycoprotein acetyls, a metabolite implicated in inflammatory diseases and cancer and associated with mortality and cardiovascular disease, and isoleucine which was previously linked to diabetes and cardiovascular disease. Increased diversity of the gut microbiota is associated with more favorable cardiovascular and metabolic profile, while imbalance in microbial diversity (lower microbial diversity) has been found in autoimmune diseases, obesity, cardiometabolic conditions (page 11, lines 205, 215, page 12, lines 231-236).

Supplementary Table 2 and 3: In the text, these data are used to support the notion that, "The direction of effect size across the cohorts was generally concordant" but this can be challenging to see across columns, and may be better represented by bar/whisker graphs with betas/SEs by cohort

and overall. It also appears that only statistically significant findings are detailed. While Supplementary Table 1 offers the list of metabolites tested, if the prior statement is correct, there is nowhere in the text or supplementary material detailing the exact microbial feature set tested. This is especially noteworthy given testing demonstrates that analyses appear to combine microbial traits of different taxonomic ranks, i.e. family with genus, which could just be due to understandable inability to resolve OTUs, or worse, the erroneous inclusion of all within the same testing set.

We have now provided plots to support the notion that the direction of effect size for statistically significant findings was generally concordant across the cohorts (Supplementary Figure 1 and 2). We have also detailed not only statistically significant but all results in the Supplementary Table 3 and 4.

We apologize for this omission. We have now provided the list of microbial taxa tested in Supplementary Table 2. With regard to the reviewer's question about use of microbial traits of different taxonomies, we would like to point out that we are not using OTUs but rather OTU independent approach. Taxonomical profiles for all the samples were done by relying on per-sequence classification to a reference database and then generating genus-level and higher taxonomical profiles directly and independently of any OTU clustering.²¹ Different taxonomical profiles generated in this way were tested independently. We have clarified this in the revised version of the manuscript on page 16, lines 314-318.

The data set should be made available as per Nature Communications guidelines.

We have now made the data availability statement more accurate in the relevant section of the revised version of the manuscript (page 17, lines 356-360). We have clarified that the summary statistics are available in Supplementary Tables 3 and 4. The Rotterdam Study data could be accessed through consultation with the management team of the cohort. Due to the new General Data Protection Regulation (GDPR), we are no longer allowed to share pseudonymized data in open or closed data repositories. The LifeLines-DEEP metagenomics sequencing data are available at the European Genome-phenome Archive under accession EGAS00001001704.

²¹ Wang, Jun, et al. "Meta-analysis of human genome-microbiome association studies: the MiBioGen consortium initiative." (2018): 101.

REVIEWERS' COMMENTS:

Reviewer #1 (Remarks to the Author):

The authors have satisfactorily responded to all my comments.

Reviewer #2 (Remarks to the Author):

In the time since their original submission, Vojinovic et al should be commended on their considerably expanded efforts to improve this manuscript. Namely, weaknesses such as relevant cohort details that were previously omitted, lack of detail regarding covariate selection and significance testing thresholds, and most importantly, the biological significance of findings have been adequately addressed. Writing overall has been tightened tremendously with more cautious interpretation, less colloquial language, and more precise declarations. Overall, they were responsive to the feedback offered by both reviewers and have also addressed questions regarding data availability.

Response to Reviewers: NCOMMS-19-09554A (Vojinovic et al.)

Reviewer #1 (Remarks to the Author):

The authors have satisfactorily responded to all my comments.

Reviewer #2 (Remarks to the Author):

In the time since their original submission, Vojinovic et al should be commended on their considerably expanded efforts to improve this manuscript. Namely, weaknesses such as relevant cohort details that were previously omitted, lack of detail regarding covariate selection and significance testing thresholds, and most importantly, the biological significance of findings have been adequately addressed. Writing overall has been tightened tremendously with more cautious interpretation, less colloquial language, and more precise declarations. Overall, they were responsive to the feedback offered by both reviewers and have also addressed questions regarding data availability.

We thank the reviewers for the in-depth-comments, suggestions, and corrections, which improved the manuscript.